# Effects of ZIF-L Morphology on PI@PDA@PEI/ZIF-L Composite Membrane’s Adsorption and Separation Properties for Heavy Metal Ions

**DOI:** 10.3390/polym15234600

**Published:** 2023-12-01

**Authors:** Hui Cao, Ziyue Jiang, Jing Tang, Qiong Zhou

**Affiliations:** 1College of New Energy and Materials, China University of Petroleum, Beijing 102249, China; candy1128a@163.com; 2The Experimental High School Attached to Beijing Normal University, Beijing 102249, China; sophiajiang40@gmail.com; 3Huakang Sub-District Office of Jinghai District People’s Government, Tianjin 301617, China; tangjinggfz@163.com

**Keywords:** ZIF-L, nanofibers, heavy metal separation membrane, heavy metal ions, filtration rate, permeability

## Abstract

Composite polymolecular separation membranes were prepared by combining multi-branched ZIF-L with high-porosity electrospinning nanofibers PI. Meanwhile, PDA and PEI were introduced into the membrane in order to improve its adhesion. The new membrane is called the “PI@PDA@PEI/ZIF-L-4” composite membrane. Compared with the PI@PDA@PEI/ZIF-8 composite membrane, the new membrane’s filtration rates for heavy metal ions such as Cd^2+^, Cr^3+^, and Pb^2+^ were increased by 7.0%, 6.6%, and 9.3%, respectively. Furthermore, the new membrane has a permeability of up to 1140.0 L·m^−2^·h^−1^·bar^−1^, and displayed a very stable performance after four repeated uses. The separation mechanism of the PI@PDA@PEI/ZIF-L composite membrane was analyzed further in order to provide a basis to support the production of separation membranes with a high barrier rate and high flux.

## 1. Introduction

Heavy metal ions, due to their low concentration, high solubility, and tendency to form complexes, are difficult to be removed/filtered from polluted water [1]. This becomes a challenging subject and has drawn some recent interest [2,3,4]. For the purpose of heavy metal ion separation, metal–organic frameworks (MOFs) have become an important research focus due to their high specific surface area and porosity, high active site, adjustable structure, and easy functionalization.

Existing studies on adsorbent materials commonly focus on the adsorption capacity of adsorbents for heavy metals in single-component heavy metal solutions; few research articles have been published regarding multi-component heavy metal solutions. In real cases of sewage treatments, however, a variety of heavy metals coexist in sewage/polluted water. It is extremely important to study the removal of adsorbents in multi-component heavy metal solutions.

Flower-shaped zeolitic imidazolate framework-L (ZIF-L) nanostructures, as MOF materials, have unique cushion cavities that can capture heavy metal ions. The free 2-methylimidazole in the cavity can also provide additional adsorption sites. ZIF-L is expected to have an excellent heavy metal ion adsorption capacity. Electrospinning is a convenient and effective method typically used to produce functional nanofibrous membranes (NFMs). The prepared NFMs have the advantages of a high porosity, large specific surface area, adjustable structure, and easy surface modification.

In the present study, a ZIF-L was anchored onto the surface of PI fibers with a PEI coating, and the synthesized PI@PDA@PEI/ZIF-L-4 composite membrane was used to separate heavy metal ions in polluted water. It is desirable to achieve an increase in water flux and the heavy metal removal rate under low pressure conditions.

## 2. Experimental Section

### 2.1. Materials

Materials, such as Pyromellitic dianhydride (PMDA), 4,4′-diamino-diphenyl ether melamine (ODA), polyethylenimide (PEI), 2-methylimidazole, zinc nitrate hexahydrate (Zn (NO_3_)_2_·6H_2_O), hydrogen peroxide (H_2_O_2_), dopamine (DA), lead acetate, cadmium nitrate, chromium trichloride, and the standard solution of disodium ethylenediamine tetraacetate, were purchased from Victrex company in Shanghai, China.

### 2.2. Preparation of ZIF-L

Based on the coordination control method, flower-shaped zeolitic imidazolate framework-L (ZIF-L) crystals were successfully synthesized at room temperature by adding H_2_O_2_ as the control agent. A 30% H_2_O_2_ solution (0, 1, 2, 3, 4, 5 mL) was added to an aqueous solution of 2-methimidazole and slowly stirred for 2 min. The obtained ZIF-L crystals were recorded as ZIF-L-0 (ZIF-8), ZIF-L-1, ZIF-L-2, ZIF-L-3, ZIF-L-4 and ZIF-L-5 [5].

### 2.3. Preparation of PI@PDA@PEI/ZIF-L Composite Membrane

As shown in Figure 1, the preparation process is divided into three parts:

**(1)** 
**Preparation of electrospun PI nanofibers**


The synthesis of polyamide acid (PAA) spinning solution displays the condensation polymerization of a low-temperature solution.

First, 30 mL N, N-dimethylformamide (DMF) solution was added to the three-mouth flasks, then, 1.8334 g of 4,4′-diaminodiphenyl ether (ODA) was added. The mixture was stirred with a mixing paddle at 400 r/min for 20 min until the ODA was dissolved. Then, the three-mouth flasks were placed in an ice-water bath, to keep the temperature of the reaction system at about 0~5 °C during the reaction. A total of 2.0121 g of terephthalate (PMDA) was added to the flask in quadruplicate (the molar ratio of PMDA to ODA was 1.01:1) with a time interval of 20 to 30 min. The whole feeding process lasted for 1.3~2 h. After adding all PMDA, the resulting solution was stirred for 2 h to obtain a clear solution of PAA with a concentration of 12 wt.%. This was kept in the refrigerator.

A volume of 2 mL of PAA solution was removed with a 2.5 mL needle syringe and fixed on the push platform of the electrospinning machine. A 30 cm wide aluminum foil was affixed to the receiving roller to receive the spinning fibers. The spinning parameters were set to the following: voltage as 16 kV, push and injection speed as 0.05 mm/min, receiving roller speed as 100 r/min, and receiving distance as 15 cm. After 12 h of spinning, PAA nanofibers with a thickness of 60 μ m were obtained.

The PAA nanofibers were placed in a maver furnace and the temperature was raised from room temperature to 300 °C within 2 h, then kept warm for 2 h and then cooled to room temperature to obtain the PI nanofiber film.

**(2)** 
**Modification of the PI nanofibers**
①A total of 0.02422 g of THP (Tetrahydropyran) was dissolved in 50 mL of deionized water to configure 4 mmol/L of the THP solution;②A total of 0.1 g HCl (hydrochloric acid) with appropriate amount of deionized water to was used to make 30 mL HCl solution;③A total of 1.5 g HCl solution (from step ②) was dropped into THP solution (from step ①) to obtain a mixed solution of PH 8.5;④A total of 0.2 g of dopamine (DA) was added to the above mixed solution (from step ③) and was settled to 100 mL; a total of 2 mg/mL of the PDA solution was obtained;⑤Two PI nanofibers of 5 × 5 cm were put into the PDA solution (mixed in step ④), and then removed after 24 h to obtain PDA@PI fibers with PDA coated on the surface.
**(3)** 
**Preparation of the PI@PDA@PEI/ZIF-L-4 composite membrane**


To obtain the PI@PDA@PEI/ZIF-L composite membrane, 5 wt.% ZIF-L-4 was added to 3 wt.% PEI solution, then we stirred and applied the above solution to the PDA@PI fiber surfaces, until the liquid completely covered the entire surface of the fibers. After drying the fiber film in an oven at 80 °C, a PI@PDA @PEI/ZIF-L membrane was obtained.

### 2.4. Characterization

The sample morphology was characterized using a scanning electron microscope (SEM, SU8010). The SEM characterization and associated element analysis were performed with a Hitachi SU8010 (Tokyo, Japan) equipped with energy-dispersive X-ray spectroscopy (EDS). Fourier transform infrared spectroscopy (FTIR, Tensor II) was used to characterize the types of functional groups and the interactions between the functional groups in a sample. The attenuated total reflection method (ATR) was used for testing of the membranes, with a scanning range from 4000 cm^−1^ to 650 cm^−1^, and the transmission method (TR) was used for testing of the powdered samples, with a scan range from 4000 cm^−1^ to 400 cm^−1^. The crystal structure was determined by performing X-ray diffraction (XRD, D8 Focus) on the sample. The test conditions were a Cu Kα target with a wavelength of 0.154 nm, a test angle of 5~70°, a voltage of 40 kV, a current of 30 mA, and a scan rate of 5°/min. The composition and structure of the sample was tested using X-ray photoelectron spectroscopy.

### 2.5. Performance Characteristics


**Heavy metal adsorption performance test**


A solution containing three heavy metals, e.g., Pb^2+^, Cd^2+^, and Cr^3+^, was prepared, and the PH of the solution was adjusted to 6 by adding diluted hydrochloric acid. A volume of 20mL of the solution was added to a certain amount of ZIF-L, and stirred at room temperature for 10 min. After the adsorption process was completed, a needle filter was used to filter the solution. The concentration of the filtered solution was measured using an inductively coupled plasma spectrometer (ICP), and the adsorption capacity and adsorption rate were calculated using the following formula:(1)$$R=\left(1-\frac{C}{C_{0}}\right)\times 100\%$$
(2)$$q=\frac{V\left(C_{0}-C\right)}{m}$$
where *C*_0_ is the initial mass concentration, mg/L; *C* is the mass concentration at the adsorption equilibrium (*C_e_*) or time t (min), mg/L; *V* is the volume of solution, L; *m* is the mass of the adsorbent, g.


**Membrane water flux test**


The permeability and barrier properties of the membranes were evaluated with a device for cross-flow filtration made by ourselves (Figure 2). This filter device is composed of two polytetrafluoroethylene plates engraved with serpentine water channels, with an effective filtration area of 6.326 × 10^−4^ m^2^. Filtration power is provided by two peristaltic pumps.

The separation membrane (to be tested) was placed into the cross-flow filter device, ensuring that the membrane completely covered the flow path, and then it was secured with screws. Firstly, the deionized water was pre-compressed for 10 min at 2 bar pressure to obtain a stable water flux. The time duration was measured for filtering out 20 mL deionized water at 1 bar water pressure. Multiple tests were then repeated, and the average value of the measured times was used to calculate the flow rate of the filtered liquid. The following formula was used to calculate the water flux of the membrane:J = v/(S × P)(3)
where J (L·m^−2^·h^−1^·MBar) is the water flux, v (L·h^−1^) is the flow rate of the filtered liquid, S (6.326 × 10^−4^ m^2^) is the effective filtration area, and P (Bar) is the pressure applied on the membrane.

## 3. Results and Discussion

### 3.1. Structure Characterization and Adsorption Properties of ZIF-L

Figure 3 shows the SEM images of the ZIF-L series crystals. As shown in Figure 3, with the addition of more H_2_O_2_, the ZIF-L gradually grows into a flower-shaped structure with elongated branches, and such branches on the crystal surface increase in number and become longer and thinner. This observation is consistent with the description in the published literature [6].

The formation mechanism of the flower-shaped ZIF-L structures can be understood as a two-step crystal growth process. Zn^2+^ first binds with 2-methyl imidazole to form the ZIF-L core, and then the H_2_O_2_ adsorbed onto the crystal core causes lattice accumulation, layer by layer, through hydrogen bonding, enabling the lateral and epitaxial growth of the crystal core. With the increase in the H_2_O_2_ concentration, the number of H_2_O_2_ molecules adsorbed onto the core surface increases, and the growth site also increases. Due to steric hindrance, the core surface tends to epitaxial growth, resulting in an increased number of elongated branches on the crystal surface.

For ZIF-L-5, in the SEM image, some broken branches were found, possibly due to the branches of ZIF-L-5 being too slender and unstable during the ultrasound process. However, no broken branches were observed in the morphology analysis of ZIF-L-3 and 4; this indicates that ZIF-L-3 and 4 of the ZIF-L structures are more stable.

In Figure 4a, ZIF-8 and ZIF-L-1, at 011, 002, 022, 013 and 222, demonstrated the successful synthesis of ZIF-8 and ZIF-L [7]. It was also noticed that ZIF-L-1 lacks the characteristic peak of ZIF-L, while ZIF-L-3, 4, and 5 are consistent with the characteristic peak of ZIF-L. This might be due to the incomplete crystallization with 1ml H_2_O_2_, not forming the characteristic ZIF-L peak. Upon adding more H_2_O_2_, the crystal structure gradually changes to ZIF-L [8], and the ZIF-L-3, 4, and 5 crystals change into a stable ZIF-L structure. This demonstrates that the addition of H_2_O_2_ to the ZIF-8 reaction system can transform the ZIF-8 crystal into ZIF-L.

In the FTIR spectrum of Figure 4b, all the ZIF-L samples display significant peaks at 756 cm^−1^, 1146 cm^−1^, 1307 cm^−1^, 1422 cm^−1^, 2920 cm^−1^, 3134 cm^−1^, and 3034 cm^−1^. The peak at 756 cm^−1^ is due to out-of-plane bending; the peaks at 1146 and 1307 cm^−1^ correspond to in-plane bending; the peaks at 1422 and 2920 cm^−1^ correspond to the expansion vibration of the C–H and C=N bonds in the imidazole ring; and the peaks at 3134 cm^−1^ correspond to the expansion vibration of the methyl group [9]. Note that a new peak appeared in the last three samples, located at 3034 cm^−1^, caused by the N–H bond, which was generated as a result of the crystal’s transition to ZIF-L. Free 2-methylimidazole is dispersed in the middle layer of the ZIF-L phase, so there are many free 2-methylimidazole (a non-crystalline molecule at the amino-binding site) [10].

The adsorption properties of the six MOF materials (i.e., ZIF-8, ZIF-L-1, ZIF-L-2, ZIF-L-3, ZIF-L-4 and ZIF-L-5) in heavy metal solutions (including 10 ppm each of Cd^2+^, Cr^3+^, and Pb^2+^) are shown in Figure 5.

As can be seen from Figure 5, the six MOF materials have excellent adsorption capacities for Cd^2+^, Cr^3+^, and Pb^2+^. The adsorption abilities are mainly attributed to the ion exchange between the ZIF-L and metal cations (cation heavy metals can replace the Zn^2+^ at its site) and the coordination between the cations and functional groups, NH and OH. In addition, electrostatic attraction also plays a synergistic role in the adsorption process [11].

The principle of the adsorption selectivity of the ZIF-L series materials to three heavy metals is similar, in the order of Cr^3+^ > Pb^2+^ > Cd^2+^. This is because high-valency ions are more likely to be adsorbed; for example, Cr^3+^ is preferentially adsorbed. ZIF-8 has a significantly higher selectivity for Cr^3+^. For Cd^2+^ and Pb^2+^, with the same valences, the adsorbent usually preferentially adsorbs divalent cations with a lower hydration energy; this is because the metal ions must separate from most of the hydration water molecules before entering the smaller channel of the adsorbent, and the lower hydration energy is more likely to escape [12]. It was reported that the Pb^2+^ hydration energy is 1425 kJ/mol and the Cd^2+^ hydration energy is 1755 kJ/mol. Clearly, Pb^2+^ has a lower hydration energy and more easily enters the micropores of the adsorbent. The heavy metal ions separated from most of the hydration water molecules are more conducive to interacting with the adsorbent’s functional groups. Therefore, Pb^2+^ is preferred over Cd^2+^ to be adsorbed [13,14,15]. This is also consistent with the experimental results of the present study.

The adsorption rates of ZIF-L-1 to ZIF-L-4 in the mixed heavy metal solution gradually increased with the addition of more H_2_O_2_. The branches from ZIF-L-1 to ZIF-L-4 structures increase in number and length, resulting in larger total surface areas in contact with the heavy metal ions, and thus, a high mass transfer efficiency. The adsorption capacities of ZIF-L-4 and ZIF-L-5 are barely similar. However, in the SEM images of ZIF-L-5, fragments and broken branches were found, and the branches were too slender and unstable; therefore, ZIF-L-4 was selected as the adsorbent for subsequent studies.

### 3.2. Characterization of PI@PDA@PEI/ZIF-L-4 Composite Membranes

Electrospinning nanofibers loaded with MOFs is a method to introduce MOFs into membranes, in which a lower separation pressure and lower energy consumption are needed, compared to the conventional mixed matrix membrane. The commonly used methods to load MOFs onto electrospinning nanofibers mainly include co-electrospinning and in situ growth; there are still certain problems such as that the active sites of the MOFs become buried or easily fall off the fiber surface.

Polyethylenimine (PEI) has many forms of amino groups, including primary, secondary, and tertiary functional amine sites. Studies have shown that PEI, with its coordination and hydrophilicity, has an excellent adsorption and chelation capacity for heavy metal ions. The present study used PEI as an adhesive to anchor the MOFs to the fibers, and improved the binding force of the PEI to PI by grafting polydopamine onto the PI nanofibers.

To obtain the PI@PDA@PEI/ZIF-8 and PI@PDA@PEI/ZIF-L nanofiber composite membranes, ZIF-8 and ZIF-L-4 were incorporated into the PEI system and coated onto PDA-modified PI fibers. Whereas nanofibers mainly provide support and promote penetration, PEI allows for the bonding effect of ZIF onto the fibers, and ZIF-L is used for the adsorption of the heavy metal ions. To explore the effect of the ZIF-L morphology on the membrane separation performance, the mechanism of membrane separation was investigated using XPS and EDS.

As shown in Figure 6a,b, the PI nanofibers have a smooth surface and a three-dimensional mesh nanostructure, and the diameter of the filament ranges from 100 to 200 nm. A handful of PDA particles were adhered to the surface of the PI fibers, resulting in an increase in the PI fiber diameters to 150~280 nm, along with a corresponding increase in the roughness of the nanofibers.

From Figure 6c,d, it can be seen that spherical particles of about 2 μm in diameter can be observed on the surface of the composite membrane mixed with ZIF-8, consistent with the morphology of the ZIF-8 crystals. The composite membrane incorporated with ZIF-L-4 (PI@PDA@PEI/ZIF-L-4 membrane) was bonded to a large number of flower-shaped ZIF-L-4 crystals with a diameter of approximately 4 μm. The incorporation of the PEI binds ZIF-8 and ZIF-L-4 to the fiber through a cross-linking network, binding ZIF-L-4 tightly to the fiber matrix and making it difficult for it to fall off.

The composition and structure of the different materials were verified by FTIR and XRD, as shown in Figure 7.

As shown from Figure 7a, in the PI@PDA@PEI membrane, distinct characteristic peaks can be observed at 3262 cm^−1^, 1569 cm^−1^, 1483 cm^−1^, and 1580 cm^−1^. The peaks at 3262 cm^−1^ and 1569 cm^−1^ belong to the bending vibrations of the NH_2_ and NH in the PEI chain, respectively. The more intense peaks at 1483 cm^−1^ and 1580 cm^−1^ represent the characteristic absorption peaks of the secondary amine and primary amine groups [16,17].

For ZIF-8 and ZIF-L-4, the peak at 756 cm^−1^ corresponds to out-of-plane bending; the peaks at 1146 and 1307 cm^−1^ correspond to in-plane bending; the peaks at 1422 and 2926 cm^−1^ correspond to the expansion vibration of the C-H and C=N bonds on the imidazole ring; and the peak at 3134 cm^−1^ corresponds to the expansion vibration of the methyl group. Note that a new peak appeared in the last three samples, located at 3034 cm^−1^, which is caused by the N-H bond. The formation of this bond is the result of the crystal transition to ZIF-L, where the free 2-methimidazole is dispersed in the middle layer of the ZIF-L phase, so there are many free 2-methimidazole (a non-crystalline molecule in the amino binding site) [6].

Compared with PI@PDA@PEI, some new peaks appeared in the PI@PDA@PEI/ZIF-8 and PI@PDA@PEI/ZIF-L-4 composite membranes. The new peak at 3136 cm^−1^ corresponds to the stretching vibration of the methyl group in ZIF-8 and ZIF-L-4, the new peak at 759 cm^−1^ corresponds to the out-of-plane bending of ZIF-8 and ZIF-L-4, and the new peaks at 1147 and 1308 cm^−1^ correspond to the in-plane bending of ZIF-8 and ZIF-L-4, demonstrating the successful loading of ZIF-8 and ZIF-L-4.

From the XRD profile in Figure 7b, the diffraction peak of ZIF-8 is consistent with those reported in previous work, where 2θ is the characteristic peaks of 7.3, 10.3, 12.7, 12.7, 14.7, 16.4, and 18.0 corresponding to the (011), (002), (112), (022), (013), and (222) surfaces [18], demonstrating the successful preparation of ZIF-8. The diffraction peak of ZIF-L-4 agrees with that of ZIF-L, as shown in the literature, indicating the successful preparation of ZIF-L-4. The peaks of both composite membranes correspond to the amorphous peaks of the polymer and the major crystallization peaks of ZIF-L. ZIF-L doping is effective, and the topology of all the species is preserved after recombination.

To further verify the structure of the composite membrane surface, an XPS analysis was performed on the PI@PDA@PEI/ZIF-L-4 composite membrane, as shown in Figure 8. The prepared composite membrane samples showed several characteristic peaks at binding energies of approximately 285.08, 397.08, and 531.8 eV, mainly corresponding to C 1s, N 1s, O 1s, and Zn 2 p. In the C1s spectrum, three main peaks can be observed at 287.7, 285.8, and 284.5 eV for the C=O, C=N, and C=C groups. Among them, the C=O group proves the existence of the PI fiber layer in the composite membrane, the C=N group is the crosslinking reaction between PEI and ECH, and the C=C group proves the presence of 2-methimidazole in PI and ZIF-L.

The high-resolution N 1s spectra (Figure 8) were fitted to two characteristic nitrogen species at 399.2 and 397.4 eV, corresponding to a neutral amine (-NH_2_/-NH) and protonated amine (-NH_3_^+^ or -NH_2_^+^), confirming the binding of PEI in the composite membrane. In addition, two characteristic peaks of Zn were displayed at 1020.6 eV (Zn2p3) and 1043.9 eV (Zn2p1). The above results showed that the PEI/ECH crosslinking system was successfully synthesized on the PI nanofibers and the ZIF-L-4 nanoparticles were successfully loaded onto the membrane surface.

### 3.3. Separation Properties of the PI@PDA@PEI/ZIF-L-4 Composite Membranes

The separation properties of the membrane were determined through permeability experiments. Through the operation of the self-made cross-flow filtration device, the water flux of the membrane was calculated using Formula (1), and the heavy metal barrier rate was calculated by measuring the concentration change in the heavy metal ions before and after the cross-flow filtration with ICP-OES. The operating pressure of the device was 1 bar, the test temperature was 25 °C, the three-component heavy metal solution was used as the feeding liquid (including Cd^2+^, Cr^3+^, Pb^2+^ each 1 ppm), and the solution flow rate was 250 mL/min.

The water flux of the PI@PDA@PEI/ZIF-8 composite membrane and PI@PDA@PEI/ZIF-L-4 composite membrane and the separation performance in the heavy metal solutions are shown in Figure 9.

In this experiment, effects of MOF morphology on the separation performances were investigated by doping the membrane with ZIF-8 and ZIF-L-4 in different appearances. As shown in Figure 9, the flower-shaped ZIF-L-4 can be more assisting to the deposition of PEI on the fiber membrane with more mosaic branching structures, and the crystal branches interlace each other, providing a long-range channel for the transport of water molecules. Therefore, the water molecules stay in the membrane for longer, and the water flux (1140 L·m^−2^·h^−1^·bar^−1^) is lower than the water flux of the ZIF-8 composite membrane (1243.44 L·m^−2^·h^−1^·bar^−1^).

As seen in Figure 9, the water flux of the PI@PDA@PEI/ZIF-L-4 composite membrane is 1140 L·m^−2^·h^−1^·bar^−1^, which is lower than such of the ZIF-8 composite membrane (i.e., 1243.44 L·m^−2^·h^−1^·bar^−1^). The fact is that, comparing with ZIF-8, the flower-shaped ZIF-L-4 (containing more mosaic branching structures and with branches between crystals cross each other) provides a long-range channel for the transport of water molecules; thus, the water molecules stay in the membrane for longer.

It can also be seen in Figure 9 that by introduction of ZIF-L-4 into the composite membrane, the removal rate of heavy metals was significantly increased. The barrier rate of Cr^3+^ and Pb^2+^ both exceed 90%. The removal rate of Cd^2+^ increased by 7%, from 73.3% to 80.4%; the removal rate of Cr^3+^ increased by 6.6%, from 90.2% to 96.8%; the removal rate of Pb^2+^ increased by 9.3%, from 90.3% to 99.6%. Due to the flower-shaped branching structure of ZIF-L-4, the total surface area of contact with heavy metals is larger than that of ZIF-8. The long-range orderly channels formed by staggered branches enforce great resistance to transmission of heavy metal ions, and also greatly improves the efficiency of heavy metal ions contacting the active site on ZIF-L-4. The adsorption effect of heavy metal ions is much better improved.

### 3.4. Reusability of PI@PDA@PEI/ZIF-L-4 Composite Membranes

In applications the practical membrane should be reusable for cost purposes. A metal complexing agent, disodium ethylenediaminetetraacetic acid, is used for repeated tests of adsorption and desorption experiments in four cycles on the PI@PDA@PEI/ZIF-L-4 (5 wt.%) composite membrane, as shown in Figure 10.

As can be seen in Figure 10, after two cycles of filtration, the membrane’s removal rates of the heavy metal ions were approximately decreased by 5%, 5%, and 20%, for Cd^2+^, Cr^3+^, and Pb^2+^, respectively. After four cycles, the removal rates for Cd^2+^ decreased from 80.53% to 73.61%; for Cr^3+^, from 96.80% to 87.74%; and for Pb^2+^, from 96.63% to 85.43%, which means that the heavy metal removal rate was relatively stable. Thus, the PI@PDA@PEI/ZIF-L-4 composite membrane can perform four filtration cycles while maintaining a stable heavy metal removal rate. This suggests that the PI@PDA@PEI/ZIF-L-4 composite membrane is reusable with stable performances.

### 3.5. Exploration of Separation Mechanism of ZIF-L/PEI/PDA@PI Composite Membrane

The composite membranes prepared in this experiment were high-flux separation membranes, permeating at low pressure. They were expected to adsorb heavy metal ions through the active sites in the membrane. The EDX mapping was inspected on the membranes, as shown in Figure 11.

As shown in Figure 11a,b, the ZIF-L-4 samples maintained their flower morphology after readsorbing the heavy metals, which confirmed its good stability. From Figure 11c, we can see that Cd^2+^, Cr^3+^, and Pb^2+^ exist on the membrane surface. It can be confirmed that the adsorption of heavy metal ions does occur on the membrane surface. When the heavy metal ions are adsorbed onto the membrane surface, the electrostatic repulsion between the incoming heavy metal ions in the solution and the adsorbed heavy metal ions will be conducive to the high degree of exclusion of the incoming heavy metal ions. The content of lead is low, probably because Pb^2+^ has a smaller hydration energy, which more easily enters the micropores of the adsorbent and also easily enters the channels of ZIF-L-4. ZIF-L-4 adsorbs Pb^2+^ through its pores. Since ZIF-L-4 has a large size and many channels, the detection depth of EDX could not reach the Pb^2+^ captured in the pores. The adsorption mechanism had to be verified by XPS.

As shown in Figure 12, the peaks of Pb4f, Cd3d, and Cr2p appeared in the spectrum of XPS after adsorption, indicating that Cd^2+^, Cr^3+^, and Pb^2+^ were successfully adsorbed onto the surface of the composite membrane. Before adsorption, the N 1s spectra were deconvoluted into two peaks at binding energies of 399.2 and 397.4 eV, corresponding to the neutral amine (-NH_2_/-NH) and protonated amine (-NH_3_^+^ or -NH_2_^+^). With the filtration contacting the heavy metal ions in the feeding solution, the N 1s spectra of the composite film at binding energies of 404.5, 400.7, and 398.73 eV were deconvoluted into three peaks. First, the new peak observed at 404.6 ev corresponds to the nitrate (-NO_3_-). The positive binding energy of the neutral and protonated amine is attributed to the binding of nitrogen atoms to heavy metal ions. Because the N atom shares lone pair electrons in the hybrid orbit coordinated between the nitrogen and heavy metal ions, coordination bonds are formed, resulting in a reduced electron cloud density, thus moving the binding energy to a high energy [19]. Before adsorption, the Zn 2p peaks were at 706.5 eV and 719.9 eV, demonstrating the presence of ZIF-L-4. After heavy metal adsorption, the Zn 2p binding energy decreased and moved negatively, indicating that the chemical environment of Zn also changed. This is mainly due to the adsorption of heavy metal ions around Zn, changing the distribution state of its electron cloud, and because Zn is also the active site of heavy metal adsorption.

## 4. Conclusions

PI@PDA@PEI/ZIF-L-4 and PI@PDA@PEI/ZIF-8 composite membranes were prepared, and the adsorption and separation performance and water flux of the two composite membranes in heavy metal ion solutions were tested and studied. The results show that, compared with the PI@PDA@PEI/ZIF-8 composite membrane, the PI@PDA@PEI/ZIF-L-4 composite membrane has a better adsorption and separation performance, and its Cd^2+^, Cr^3+^, and Pb^2+^ removal rates were increased by 7.0%, 6.6%, and 9.3%, respectively; the permeation flux reached 1140.0 L·m^−2^·h^−1^·bar^−1^. The PI@PDA@PEI/ZIF-L-4 composite membrane’s performance remained stable after four repeated uses.

## Figures and Tables

**Figure 1 polymers-15-04600-f001:**
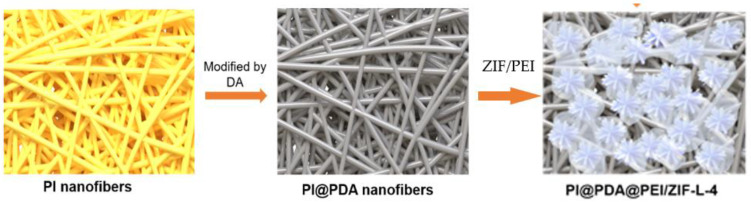
Schematic diagram of preparation of PI@PDA@PEI/ZIF-L composite membrane.

**Figure 2 polymers-15-04600-f002:**
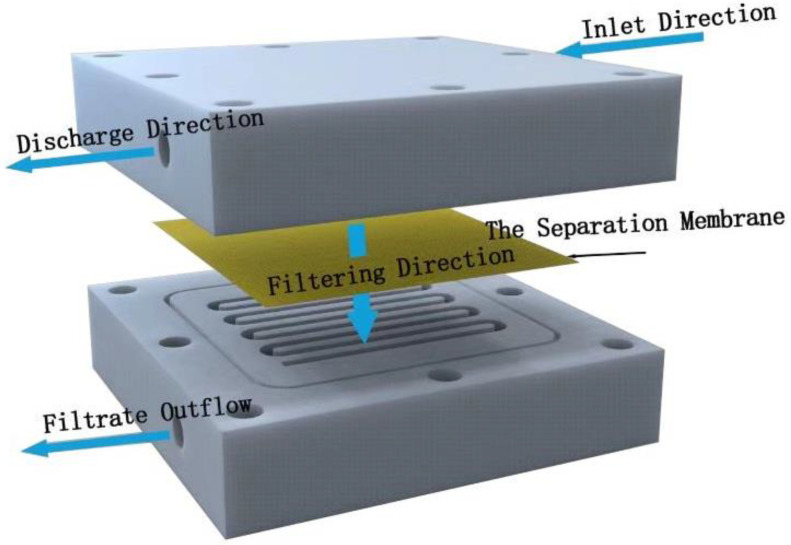
Schematic diagram of self-made experimental device for cross-flow filtration.

**Figure 3 polymers-15-04600-f003:**
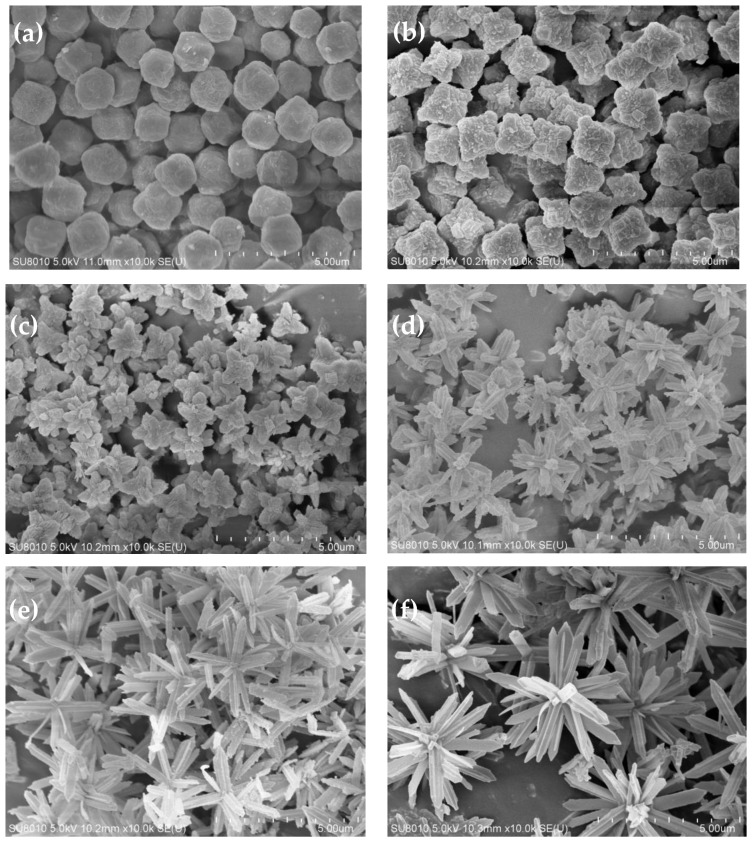
SEM images of ZIF-L crystals obtained when adding (**a**) 0 mL, (**b**) 1 mL, (**c**) 2 mL, (**d**) 3 mL, (**e**) 4 mL, and (**f**) 5 mL H_2_O_2_.

**Figure 4 polymers-15-04600-f004:**
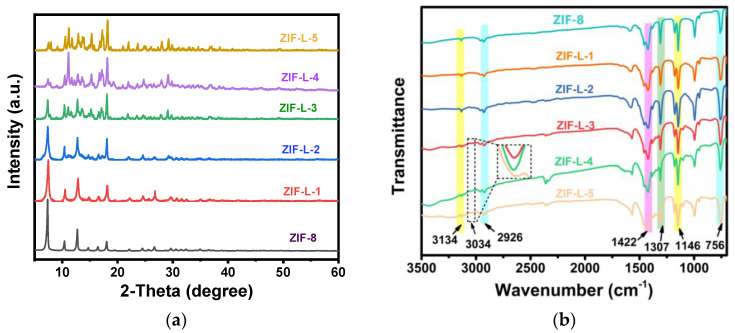
The XRD and FTIR characterization plots of ZIF-L (**a**) XRD, (**b**) FTIR.

**Figure 5 polymers-15-04600-f005:**
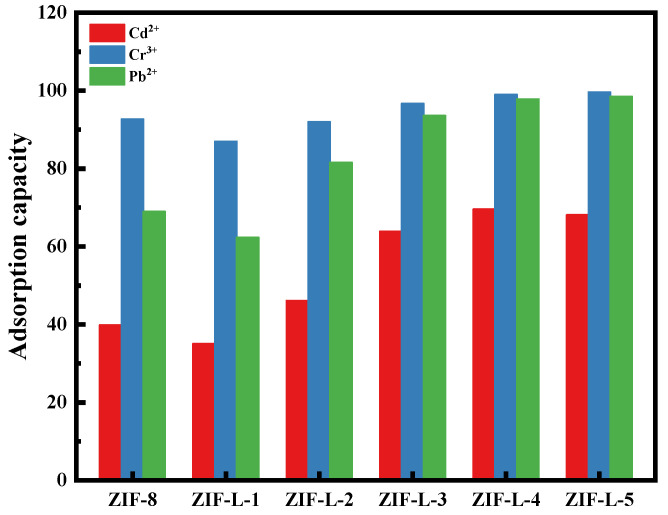
Adsorption of six MOF materials in mixed heavy metal solutions. (including 10 ppm of Cd^2+^, Cr^3+^, Pb^2+^).

**Figure 6 polymers-15-04600-f006:**
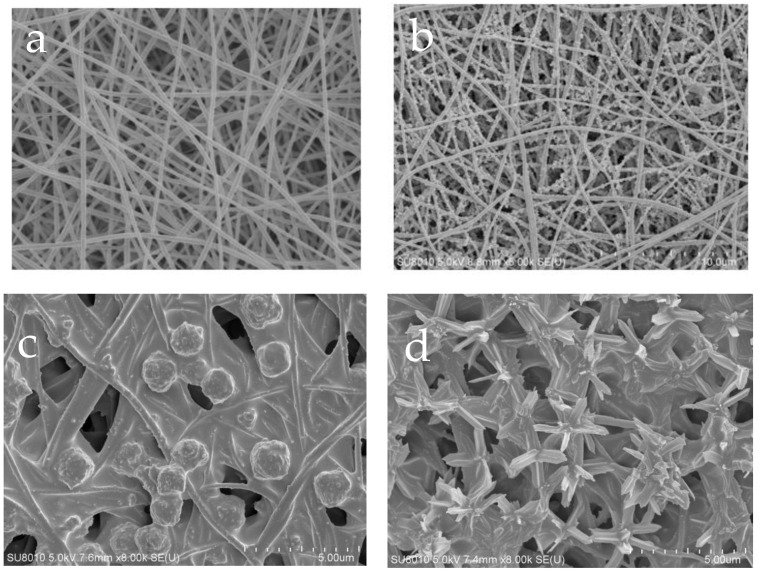
SEM images of different membranes. (**a**) PI nano-fiber, (**b**) PI@PDA membrane, (**c**) PI@PDA@PEI/ZIF-8 membrane, (**d**) PI@PDA@PEI/ZIF-L-4 membrane.

**Figure 7 polymers-15-04600-f007:**
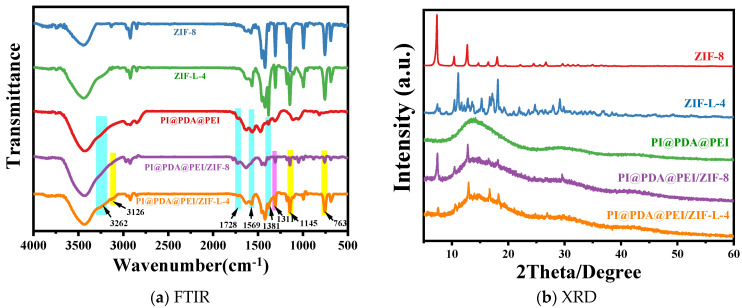
(**a**) FTIR spectra and (**b**) the XRD curve of each component.

**Figure 8 polymers-15-04600-f008:**
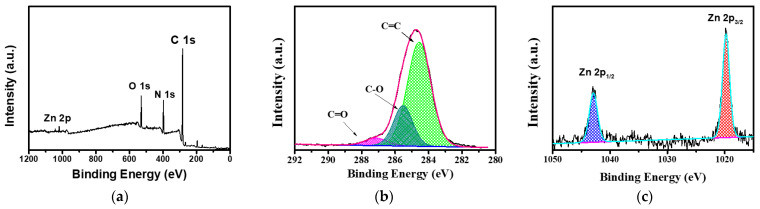
XPS spectra of PI@PDA@PEI/ZIF-L-4 composite films. (**a**) Total spectra, (**b**) C 1s spectra, (**c**) Zn 2p spectra.

**Figure 9 polymers-15-04600-f009:**
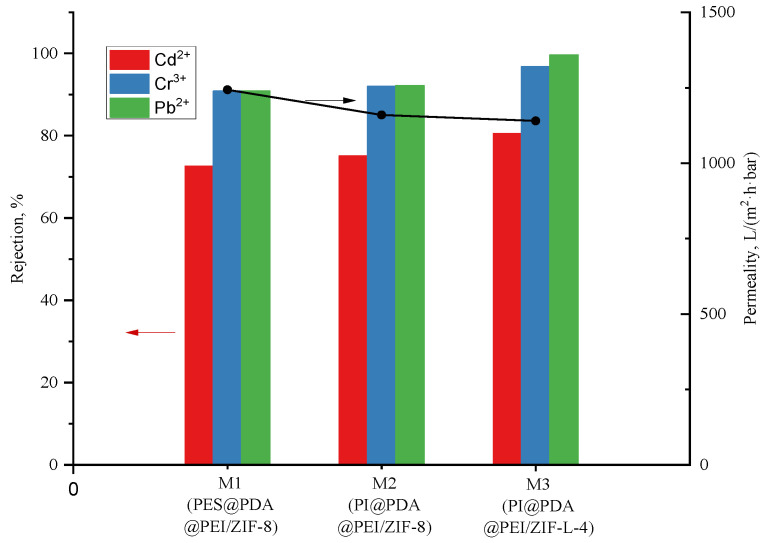
Water flux and heavy metal rejection of PI@PDA@PEI/ZIF-8 membrane and PI@PDA@PEI/ZIF-L-4 membrane.

**Figure 10 polymers-15-04600-f010:**
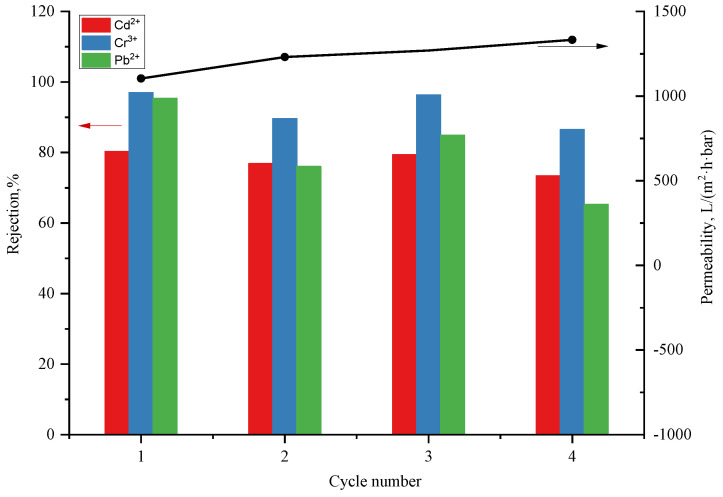
Heavy metal rejection and water flux of PI@PDA@PEI/ZIF-L-4 membrane during four filtration-washing cycles.

**Figure 11 polymers-15-04600-f011:**
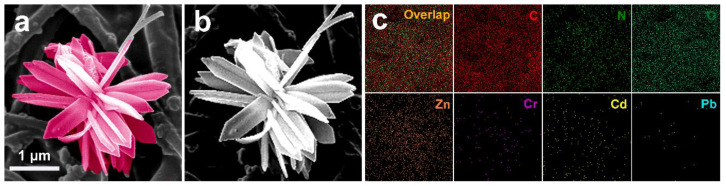
EDX mapping of the surface of the PI@PDA@PEI/ZIF-L-4 membrane after heavy metal adsorption. (**a**,**b**), ZIF-L-4 samples maintained flower morphology after readsorption bing heavy metals, which confirmed its good stability; (**c**) Cd2+, Cr3+ and Pb2+ exist on the membrane surface.

**Figure 12 polymers-15-04600-f012:**
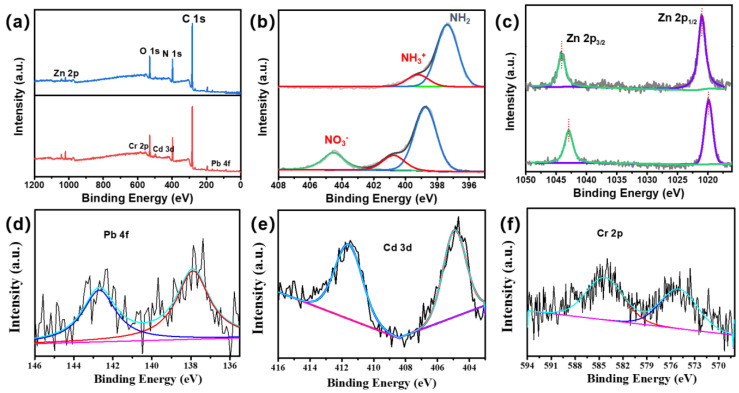
XPS spectra of (**a**) total (**b**) N 1s (**c**) Zn 2p (**d**) Pb 4f (**e**) Cd 3p (**f**) Cr 2p on the membrane surface before and after PI@PDA@PEI/ZIF-L-4 membrane adsorption.

## Data Availability

The data are contained within the article.

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
