# Peer review of "Effects of ZIF-L Morphology on PI@PDA@PEI/ZIF-L Composite Membrane’s Adsorption and Separation Properties for Heavy Metal Ions"

_polymers, 2023, doi:10.3390/polym15234600_

Round 1

Reviewer 1 Report

Comments and Suggestions for Authors

Line 117-113 I suggest providing the dimensions of the filtration chamber.

Line 117-113 Did the serpentine water channels used limit the effective filtration surface? How densely spaced were the water channels?

Fig.3 and line 144-151   I propose that in the description, both under the figure and in the text, we should use not the volume of added H2O2 but its concentration in the solution in which the ZIF-L crystals grew. Alternatively, I suggest providing the H2O2 concentration value obtained after adding a specific volume of H2O2 in the text. The amount of H2O2 added alone, regardless of its concentration and the volume of the solution to which it was added, does not contain information about the concentration of H2O2 in the solution, except that the concentration was getting higher. Although the concentration of H2O2 in the solution can be calculated based on the description in the methodology, this information should be provided by the authors of the publication.

Line 166 I suggest specifying the adsorption capacity axis unit in Fig. 5.

Line 193-199 What type of membranes can the obtained membranes be classified as (microfiltration, ultrafiltration?)

Line 193-199 Has the pore size of the membrane been determined or estimated?

Lines 296-298 How much water was filtered during flow measurements and did the water flow change during filtration in a single filtration cycle?

Line 313 In Figure 9, I suggest supplementing the legend with water flow.

Line 316-318 How was regeneration performed and were the membranes regenerated after each cycle?

Line 316-318 How much solution was filtered in each filtration cycle?

Line 316-327 W There is no description of how the water flow changes after each filtration cycle.

Line 329 In Figure 10, I suggest supplementing the legend with water flow.

Author Response

The articles in the attachment have been revised as recommended in the revised draft, please kinldy refer to the attahced file for our replies to the comments.

Reviewer 2 Report

Comments and Suggestions for Authors

The authors have developed a "PI@PDA@PEI/ZIF-L-4" composite membrane, demonstrating its remarkable stability even after four repeated uses. The study also delved into understanding the separation mechanism of these membranes, paving the way for the production of separation membranes with high barrier rates and flux. While the research was well-conducted, there are specific areas that need improvement, outlined below:

The full name of PI should be clearly stated.

The explanation for Figure 1 was insufficient. Please provide a detailed explanation.

The paragraph following Figure 11 lacked clarity. The differentiation between Cd, Cr, or Pb in Figure 11 is not evident, and the explanation needs to be improved. Consider either enhancing the image to clearly show the differences between Cd, Cr, and Pb in Figure 11 or revising the corresponding explanation to address this.

Author Response

(The authors gave the same response as above.)

Round 2

Reviewer 2 Report

Comments and Suggestions for Authors

The authors provided transparent and satisfactory responses to the reviewers' queries and carried out effective revisions to the manuscript. I recommend the publication of this revised version of the manuscript.